# Fibroblast growth factor 2 inhibits myofibroblastic activation of valvular interstitial cells

**Marcus Ground**[1]*, **Steve Waqanivavalagi**[2,3], **Young-Eun Park**[3], **Karen Callon**[3], **Robert Walker**[1], **Paget Milsom**[2], **Jillian Cornish**[3]

**1** Department of Medicine, Dunedin School of Medicine, University of Otago, Dunedin, New Zealand,
**2** Green Lane Cardiothoracic Surgery Unit, Auckland City Hospital, Auckland District Health Board, Grafton,
New Zealand, **3** Department of Medicine, Faculty of Medical and Health Sciences, University of Auckland,
Grafton, New Zealand

* groma788@student.otago.ac.nz

pone.0270227

Hospital Medical Center, UNITED STATES

**Data Availability Statement:** Data cannot be
shared publicly because of adherence to Auckland
Health Research Ethics Committee principles of
data stewardship. Our data must be stored

## Abstract

Heart valve disease is a growing problem worldwide. Though very common in older adults,
the mechanisms behind the development of the disease aren't well understood, and at pres-
ent the only therapeutic option is valve replacement. Valvular interstitial cells (VICs) may
hold the answer. These cells can undergo pathological differentiation into contractile myofi-
broblasts or osteoblasts, leading to thickening and calcification of the valve tissue. Our
study aimed to characterise the effect of fibroblast growth factor 2 (FGF-2) on the differentia-
tion potential of VICs. We isolated VICs from diseased human valves and treated these
cells with FGF-2 and TGF-β to elucidate effect of these growth factors on several myofibro-
blastic outcomes, in particular immunocytochemistry and gene expression. We used TGF-β
as a positive control for myofibroblastic differentiation. We found that FGF-2 promotes a
'quiescent-type' morphology and inhibits the formation of α-smooth muscle actin positive
myofibroblasts. FGF-2 reduced the calcification potential of VICs, with a marked reduction
in the number of calcific nodules. FGF-2 interrupted the 'canonical' TGF-β signalling path-
way, reducing the nuclear translocation of the SMAD2/3 complex. The panel of genes
assayed revealed that FGF-2 promoted a quiescent-type pattern of gene expression, with
significant downregulations in typical myofibroblast markers α smooth muscle actin, extra-
cellular matrix proteins, and scleraxis. We did not see evidence of osteoblast differentiation:
neither matrix-type calcification nor changes in osteoblast associated gene expression were
observed. Our findings show that FGF-2 can reverse the myofibroblastic phenotype of VICs
isolated from diseased valves and inhibit the calcification potential of these cells.

## Introduction

Heart valve disease is highly prevalent globally [1], and as populations age, the incidence of
aortic stenosis and end-stage calcific aortic valve disease (CAVD) is set to drastically increase
[2]. At present, the only therapeutic option for treating these diseases is replacement of the

securely at the University of Auckland. Data are available from the University of Auckland on request by emailing humanethics@auckland.ac.nz (https://www.auckland.ac.nz/en/research/about-our-research/human-ethics/ahrec.html).

**Funding:** MG, SW, PM and JC received funding from the Green Lane Research and Educational Fund, grant number 19/43/4145, https://www.greenlaneresearch.co.nz/ The funders had no role in study design, data collection and analysis, decision to publish, or preparation of the manuscript.

**Competing interests:** The authors have declared that no competing interests exist

valve with a prosthesis [3]. Despite its prevalence, not much is known about the molecular mechanisms behind the development of heart valve disease. Gone are the days when age-related disease such as CAVD were described as simple 'wear and tear' [4–6]. Instead, recent research has pointed to dysregulation of resident cells as the likely culprit [7–9]. Valvular inter-stitial cells (VICs) are the fibroblasts that populate the interstitium of the valve. Aortic valves are composed of three distinct layers: the collagen-rich fibrosa on the aortic side, the elastin-rich ventricularis underneath, and the GAG-rich spongiosa sandwiched between [10]. VICs are a heterogenous cell population, and have distinct phenotypes in keeping with the different functions of the valve layers [11]. Surrounding the valve is a monolayer of valvular endothelial cells (VECs), also distinct from endothelial cells found elsewhere in the vasculature [12]. Gene expression profiles of VICs and VECs differ from valve to valve, from leaflet to leaflet, and even from side to side of the same leaflet [13–15]. The VICs and VECs work in tandem to maintain the integrity and extracellular matrix (ECM) homeostasis of the valve over the life-time of the heart, some 3 billion beats [8, 16]. Previous research has highlighted the role of transforming growth factor β (TGF-β) stimulated 'activation' of VICs into myofibroblasts, termed 'activated VICs' or aVICs in the context of the heart valve [17, 18]. Myofibroblasts are a contractile phenotype of fibroblasts implicated in matrix production during wound healing. In health, these cells maintain tissue homeostasis, but excessive myofibroblastic activation is responsible for a host of fibro-contractile diseases [19]. In the aortic valve, this presents initially as thickening of the valve leaflet (aortic sclerosis) followed by progressive calcification leading to end-stage CAVD [20]. A thorough review of VIC physiology by Rutkovskiy et al. describes the bi-potent differentiation potential of these cells into aVICs or osteoblasts [16]. Both are responsible for valve calcification, albeit through two distinct mechanisms [8]. TGF-β is a pow-erful stimulator of myofibroblast activation through a number of biochemical pathways, in particular the canonical and non-canonical SMAD pathways [16, 21]. Recently, evidence has emerged that fibroblast growth factor 2 (FGF2) opposes this effect [22–24]. However, this evi-dence base is founded on either porcine cells [22, 23] or cells isolated from healthy human donors [24]. What remains unclear is if these anti-myofibroblast effects are able to halt or reverse pathological changes in already diseased valve cells. FGF-2 has also been shown to inhibit osteoblastic-type calcification of porcine VICs via the Notch1 pathway [25]. This study aimed to elucidate the effect of FGF-2 on key outcomes relating to 1) myofibroblastic activa-tion, and 2) calcification of VICs from diseased patients. We aimed to compare the observed changes in FGF-2 stimulated cells to TGF-β stimulated VICs, which can be thought of as a pos-itive control for myofibroblastic activation. In particular, our study aimed relate myofibroblas-tic changes to changes in gene expression. We assayed VICs isolated from patients with end-stage CAVD (a cell population displays a high degree of myofibroblastic activation from the outset), and any improvement in outcomes would constitute an *in vitro* therapeutic effect. We hypothesised that FGF-2 would oppose the myofibroblastic effect demonstrated in the TGF-β group in all areas and reverse the calcification potential of diseased VICs compared to control. We also hypothesised that osteoblastic differentiation would be inhibited by FGF-2 and pro-moted by TGF-β.

## Materials and methods

### hVIC isolation and expansion

Ethical approval was obtained for the use of human tissue from Auckland Health Research Ethics Committee (approval #AH1186). All participants provided written consent prior to tis-sue collection. 52 aortic valves were obtained from patients undergoing surgical valve replace-ment at Auckland City Hospital. All patients had end-stage CAVD. Explanted valve leaflets

were agitated in phosphate buffered saline (PBS) containing 1000U/mL collagenase type II (Sigma) for 10 minutes at 37˚C to loosen the endothelium. The suspension resulting from this first digest was centrifuged and cultured separately to identify endothelial cells. Leaflets were then scraped with a scalpel blade to remove remaining endothelial cells, then rinsed in PBS. Calcified portions of the valve were removed, and the remaining tissue was finely minced. The tissue was further digested in 1000U/mL collagenase for 2 hours at 37˚C under agitation. The resulting suspension was passed through a 70μm cell strainer to remove debris, centrifuged, and resuspended in Dulbecco's modified Eagle medium (DMEM) supplemented with 10% fetal bovine serum (FBS). Cells were passaged at 80% confluency and expanded to yield at least $1x10^6$ cells per patient. All cultures were used between passage 2 and 5.

## Experimental conditions

hVICs were cultured in 3 media formulations:

- 'Control' medium: DMEM supplemented with 10% FBS

- 'FIB' medium, previously described by Latif and colleagues [24]: DMEM supplemented with 2% FBS, 10ng/mL FGF-2 (Thermo Fisher) and 50ng/mL insulin (Sigma). Per Latif et al., the low serum content was used to reduce the inherent concentration of TGF-β present in FBS, and the insulin was added as mitotic agent

- 'TGF-β' medium: DMEM supplemented with 10% FBS and 20ng/mL TGF-β (Thermo Fisher). This media was our 'positive control' for aVIC differentiation

In all assays, hVICs from a single donor were initially seeded in control medium and allowed to adhere for 24 hours. Control medium was then aspirated and replaced with the one of the above formulations. Unless otherwise stated, endpoints were measured on day 2 and 4.

## alamarBlue cell viability assay

$7.5x10^3$ hVICs were seeded in 950μL medium, in a 24-well plate with replicates of 4. Negative control wells contained each of the corresponding media without cells. At time = 0h, 24h, 48h, 72h, and 96h, 50μL of alamarBlue cell viability reagent (Thermo Fisher) was added to each well (final concentration = 5%) and incubated at 37˚C for 4 hours. Media was aspirated and added to a 96-well plate in triplicate. Fluorescence was measured at 530/590nm on a Synergy 2 microplate reader (Biotek). Absolute fluorescence values for 4 biological repeats were combined and are presented graphically normalised to each well on day 0. An ordinary two-way ANOVA was used to compare treatments to control at each time point. To investigate relative rates of proliferation, a simple linear regression model was used to determine the gradient of the line of best fit from 24hrs onwards. The 0h data point was excluded from the linear regression analysis because fluorescence values were not comparable in the FIB group; the change in serum content after the 0hrs caused inherent differences in fluorescence of the media. Gradients were normalised to control in each replicate, and a one-way ANOVA was used to compare the two treatment groups to control.

## Immunocytochemistry

$1x10^4$ hVICs were seeded in 1mL medium in 24-well plates and cultured for 2 and 4 days. Media were changed on day 2. At the designated endpoints, wells were washed twice in PBS and fixed in 4% paraformaldehyde at room temperature for 20 minutes. hVICs were permeabilised in 0.1% v/v Triton X100 in PBS for 15 minutes, before rinsing three times in PBS. Cells were then blocked in PBS supplemented with 5% serum from the species in which the

secondary antibody was raised. Five immunocytochemistry (ICC) groups were used, detailed in Table 1. The vimentin/α-smooth muscle actin (α-SMA) protocol was used to measure cell morphology data (area, major and minor axes, aspect ratio, and circularity), as well as α-SMA staining intensity. The von Willebrand factor protocol was used to ensure the cultures were free from endothelial cells. In this protocol, endothelial cell culture obtained from the first collagenase digest was used as a positive control. Paxillin staining was used to resolve and count the number of focal adhesions per cell. SMAD2/3 staining was used to identify nuclear localisation of the phosphorylated SMAD2/SMAD3 complex. Scleraxis staining was used to identify the intracellular location of scleraxis, a transcription factor associated with tendon development and cardiac myofibroblasts.

ICC images were captured at 40x magnification on an Olympus CKX41 microscope fitted with an Olympus DP74 camera and processed using ImageJ. For the vimentin/α-SMA protocol, images were taken from n = 6 biological replicates. For each timepoint, 6 images were taken per replicate for a total of 36 images per experimental group. Images were thresholded in the green channel (vimentin) and cell morphology data was obtained from cell outlines. Cells were excluded from analysis if they overlapped other cells, or were undergoing mitosis (i.e., 2 visible nuclei), otherwise all visible cells were counted. For the paxillin stained cells, images were taken from n = 4 biological replicates. For each time point, 10 images were taken per replicate for a total of 40 images per experimental group. Images were thresholded and outlined, before being 'despeckled' to remove artefacts and non-specific staining. Mature focal adhesions were then counted manually. For SMAD2/3 staining, n = 3 biological repeats were imaged on day 2. For scleraxis staining, n = 3 biological repeats were stained on day 2 and 4.

## Alizarin red calcification stain

hVICs (n = 4) were seeded, cultured, and fixed as described in the ICC section. 2% w/v alizarin red in diH$_2$O was passed through a 0.2μm filter and adjusted to pH 4.1 with 12M HCl. 2 wells per replicate were incubated in the alizarin red for 15 minutes at room temperature before

**Table 1. Immunocytochemistry (ICC) protocols.**

| ICC group | Primary antibody | Protocol | Secondary antibody | Protocol | Counterstain | Protocol |
|---|---|---|---|---|---|---|
| Vimentin and α-SMA | Vimentin, rabbit anti-human (abcam, 92547) | 1/800 dilution, incubated for 12 hours at 4˚C | Donkey anti-rabbit IgG Alexa Fluor 488 conjugated, (Invitrogen, A32790) | 1/400 dilution, incubated for 1 hour at room temperature | DAPI | 2μg/mL, incubated for 5 minutes at room temperature |
| | A-SMA, Cy3 conjugated, mouse anti-human (Sigma C6198) | 1/200 dilution, incubated for 12 hours at 4˚C | n/a (directly conjugated) | | | |
| Von Willebrand factor (vWF) | vWF rabbit anti-human (abcam, 6994) | 1/200 dilution, incubated for 12 hours at 4˚C | Donkey anti-rabbit IgG Alexa Fluor 488 conjugated, (Invitrogen, A32790) | 1/400 dilution, incubated for 1 hour at room temperature | DAPI | 2μg/mL, incubated for 5 minutes at room temperature |
| | | | | | Phalloidin | 1/50 dilution, incubated for 1 hour at room temperature |
| Paxillin | Paxillin, mouse anti-human (Invitrogen, MA513356) | 1/50 dilution, incubated for 12 hours at 4˚C | Goat anti-mouse IgG Alexa Fluor 488 conjugated, (abcam, ab150113) | 1/400 dilution, incubated for 1 hour at room temperature | none | |
| SMAD2/3 | Activated SMAD2/3 complex, rabbit anti-human (Invitrogen, PA5110155) | 1/100 dilution, incubated for 12 hours at 4˚C | Donkey anti-rabbit IgG Alexa Fluor 488 conjugated, (Invitrogen, A32790) | 1/400 dilution, incubated for 1 hour at room temperature | Phalloidin | 1/50 dilution, incubated for 1 hour at room temperature |
| Scleraxis | Scleraxis, mouse anti-human (Santa Cruz Biotech, 518082) | 1/100 dilution, incubated for 12 hours at 4˚C | Goat anti-mouse IgG Alexa Fluor 488 conjugated, (abcam, ab150113) | 1/400 dilution, incubated for 1 hour at room temperature | Phalloidin | 1/50 dilution, incubated for 1 hour at room temperature |

washing 5 times with PBS. Calcific nodules were counted manually, and representative images were captured at 40x magnification.

## Gene expression

$5 \times 10^4$ hVICs were seeded in 6-well plates in 2.5mL medium and cultured for 2 and 4 days. Cells were harvested by adding 600μL RLT lysis buffer (Qiagen) supplemented with 1% v/v β-mercaptoethanol and dissociated from the plastic using a cell-scraper. Lysates were homogenised by passing through a 22-gauge needle 5 times. RNA was extracted using the RNeasy mini kit (Qiagen) according to the manufacturer's protocol. Genomic DNA was depleted on-column using DNase I (Qiagen). RNA concentration and purity was assessed using Nanodrop. RNA integrity was assessed using RNA screen tape (Agrilent). 500ng of RNA was reverse transcribed using the SuperScript III system (Invitrogen), modified with 100mM RNaseOUT RNase inhibitor (Invitrogen). Real-time quantitative polymerase chain reaction (RT-qPCR) was performed on the samples using Taqman probes detailed in S1 Table. Briefly, cDNA, probes, and mastermix were added to a 384-well plate in triplicate in accordance with the manufacturer's protocol and cycled on a Quantstudio 5 real-time PCR machine. 18S was used as the housekeeper gene. Genes were considered 'not detectable' if mean Ct values exceeded 37. Gene expression changes were calculated using the $2^{-\Delta\Delta Ct}$ (Livak) method [26], where both FIB and TGF-β groups were compared to the control group at day 2. Gene expression results are reported as 'fold change' and are presented graphically as $\log_2$(fold change). RT-qPCR experiments were performed twice to ensure reproducibility; presented data is from the first replicate.

## Statistics

Data were processed using Prism (GraphPad). For semiquantitative analysis of immunocyto-chemistry data (cell size and shape parameters), an ordinary 2-way ANOVA with Šidák correction was used to determine inter-group and inter-time point differences. The two experimental groups were compared to the control group within each time point. For gene expression data, a mixed effects 2-way ANOVA was applied to the ΔΔCt values. Each experimental group was compared to the control group within the same time point. Across all statistical models, differences between experimental group versus control were considered significant if $p < 0.05$, and are marked with *, differences where $p < 0.01$ are marked with #.

# Results

## Cell culture and gene expression: quality control

Of the 52 aortic valves obtained, 40 resulted in successful cell populations. Valves that were heavily calcified, or from individuals >80 years of age typically did not yield enough viable cells to form a culture. We used vWF immunostaining to ensure absence of valvular endothelial cells (hVECs) in culture (Fig 1). As a positive control, cells obtained from the first collagenase digest were plated and stained alongside the VIC cultures. The hVEC positive control culture showed strong presence of vWF, while the 'VIC' only cultures were negative for vWF. Measures of nucleic acid purity and integrity are available in S2 Table.

## FIB media is pro-proliferative

hVICs treated with FIB medium containing FGF-2 caused a significant increase in cell viability, indicative of an increase in cell number (Fig 2A). Marked increase in fluorescence from time 0h to 24h can be explained by inherent differences in the fluorescence of media

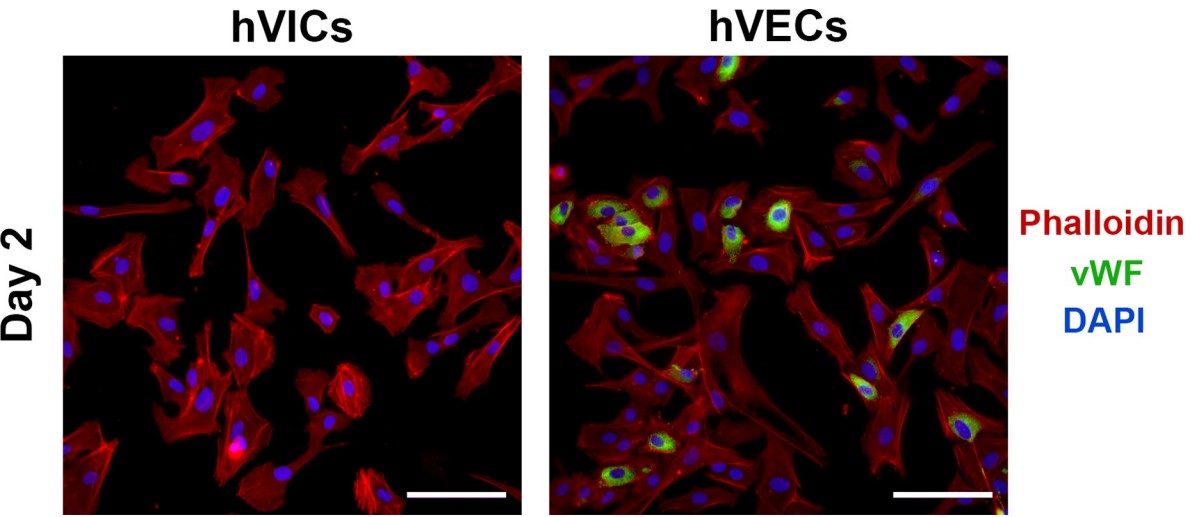

**Fig 1. vWF immunostaining.** hVICs and hVECs stained for vWF (green), phalloidin (red), and DAPI (blue). Scale bars represent 100μm.

containing different serum concentrations, and so time = 0h was excluded from linear regression analysis. Lines of best fit were added to fluorescence values of each replicate independently, with a mean $r^2$ value of 0.895 ($\sigma^2$ = 0.082). Absolute gradient values varied substantially between repeats, and so each repeat was normalised to the gradient of the control line. Gradients in the FIB group were significantly greater than in control (p = 0.0057), indicating that hVICs grown in this medium proliferated at a higher rate. TGF-β did not influence viability. A limitation of the linear regression method is it assumes cells proliferate in a linear fashion, which in practice they don't [27]. However, we believe assessing the rate of change over three 24-hour periods is sufficiently short to apply a linear fit.

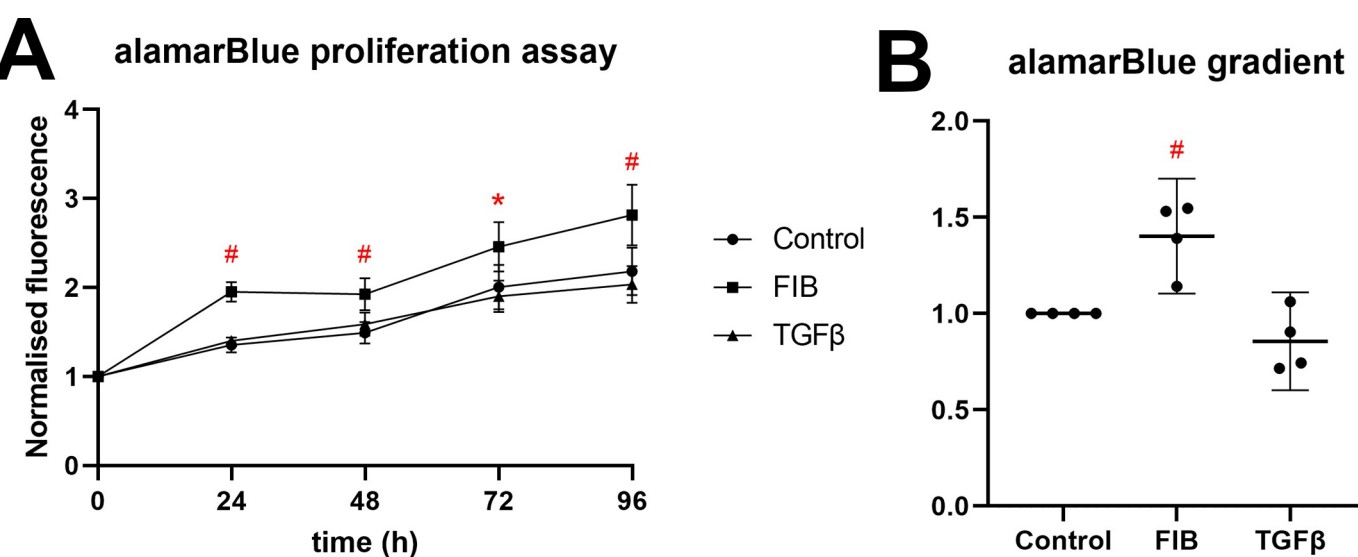

**Fig 2. FIB media is pro-proliferative.** (A) Fluorescence data from alamarBlue proliferation assay (n = 4), normalised to day 0. The two treatment groups were compared to control at each time point using an ordinary 2-way ANOVA and (B) the gradients of the line of best fit derived from a simple linear regression model of 4 repeats, normalized to control. Both treatments were compared to control using an ordinary one-way ANOVA. * p<0.05, #p<0.01.

## FGF-2 promotes quiescent-like morphology

hVICs displayed variable morphologies depending on the media formulation. Cells seeded in control and FIB media tended to grow separately or on top of each other, while cells in TGF-β supplemented media formed extensive cell-cell junctions, and in areas of high density formed a monolayer sheet (Fig 3A). Cells in the control group showed the most variability in their shape and growth pattern, while cells in the FIB and TGF-β groups tended to display similar characteristics across repeats: namely, FIB cells were slender, spindle-shaped, with small bodies and long processes, and TGF-β cells were uniformly very large and round. The variable that differed most markedly between groups was cell area. FIB-media hVICs over both time points measured, on average, 2134 μm$^2$ compared to 3238 μm$^2$ in the control group, representing a 35% reduction (Fig 3B). TGF-β treated hVICs on the other hand were more than double the size of control cells, with a mean area of 7943 μm$^2$. FIB-treated cells had a significantly higher aspect ratio (Fig 3C, p = 0.0014) and significantly lower circularity (Fig 3D, p<0.0001), indicating the cells were more spindle-like in shape. In both these variables, TGF-β treated cells did not differ from control. Finally, the length of the major axis was somewhat reduced in the FIB group (Fig 3E, p = 0.0218), but markedly increased in the TGF-β group (p<0.0001). Interestingly, 2-way ANOVA did not find any significance in the 'time factor', indicating no significant change between time points in any of the morphology variables measured.

## FGF-2 inhibits α-SMA expression

α-SMA is a well-characterised marker of aVICs [17, 22, 24, 28]. Immuno-staining (Fig 3A) revealed heterogenous expression of α-SMA in the control group on day 2, with approximately half the cells showing some degree of expression, indicating a high degree of inherent

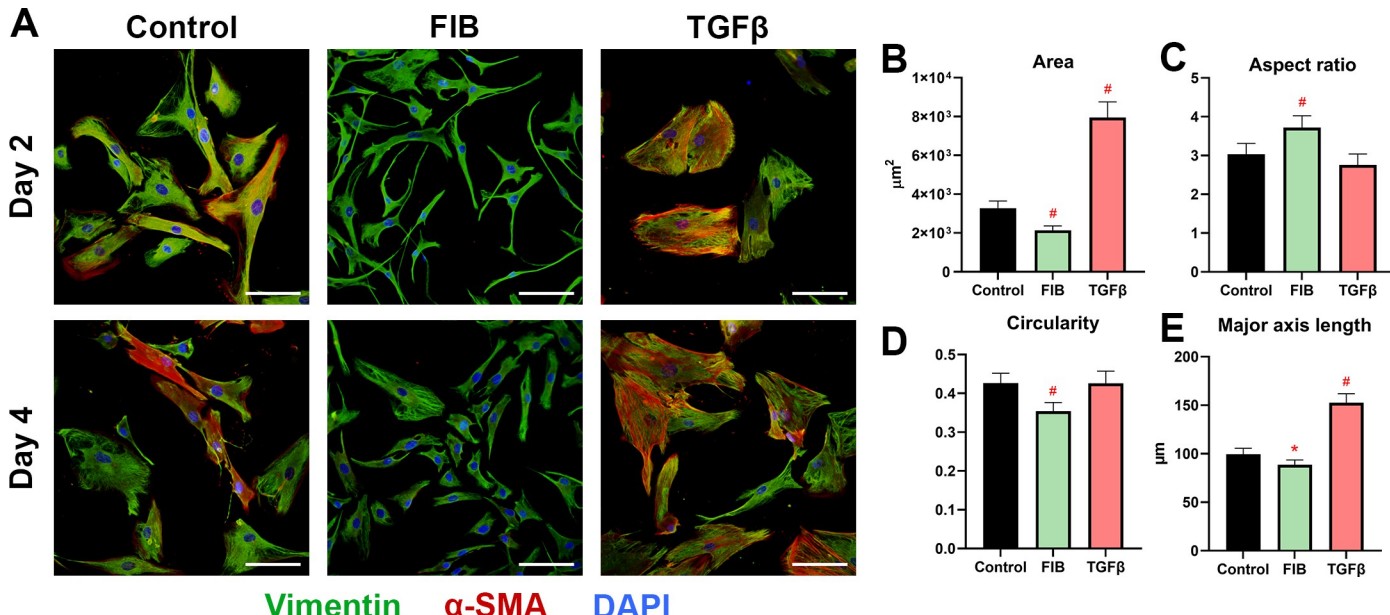

**Fig 3. FGF-2 and TGF-β differentially influence cell morphology.** (A) depicts representative images of hVICs on day 2 and 4, immunostained for α-SMA (red), vimentin (green), and DAPI (blue). Scale bars represent 100μm. The following morphological characteristics are derived from pooling days 2 and 4, as the 2-way ANOVA showed that the 'time' factor did not significantly affect any parameter. Comparisons are ordinary one-way ANOVAs between control and the experimental groups. For image analysis, 6 biological replicates were used (n = 6); 6 representative images were taken from each biological replicate per treatment per timepoint, representing >75 cells per experimental group. These are: (B) cell area in μm$^2$, (C) aspect ratio, the long axis divided by the short axis of each cell, (D) the 'circularity' of each cell. This measure is defined by $4\pi(area/perimeter^2)$, and E) the length of the major axis. * p<0.05, #p<0.01, error bars represent 95% CI.

activation. By day 4, thick bands of α-SMA were present in the majority of control cells. In the FIB group, over both time points, α-SMA staining was exceedingly low. The TGF-β treated hVICs, however, expressed large amounts of α-SMA over both time points, organised into thick, mature bands that extended across cell boundaries, linked by focal adhesions. The apparent down- and up-regulation of α-SMA in FIB and TGF-β groups respectively is supported by the RT-qPCR results (Fig 4). FIB-treated hVICs showed 7.3-fold reduction in *ACTA2* expression on day 2 (95% CI: 5.2–12.3), and a 7.9-fold reduction on day 4 (95% CI: 3.5–112) when compared to day 2 control. Conversely, TGF-β treated hVICs demonstrated a 3.5-fold upregulation on day 2 (95% CI: 2.1–5.0) and 2.5-fold upregulation on day 4 (95% CI: 0.2–4.7) compared to control. Vimentin is a cytoskeletal protein present in all VIC phenotypes, and perhaps unsurprisingly, staining was uniform across treatment groups and time points (Fig 3A). The FGF-2 treated cells did show a small but significant upregulation in *VIM* expression at day 2 and 4, while TGF-β had no effect (Fig 4). Our findings indicate that α-SMA and its corresponding gene *ACTA2* remain useful markers of TGF-β stimulated activation of hVICs. FGF-2 treatment reduced visible immune-staining of α-SMA and expression of *ACTA2*.

### FGF-2 inhibits nuclear translocation of SMAD2/3

The canonical TGF-β signalling pathway involves the phosphorylation and complex formation of SMAD2 and SMAD3, followed by the migration of this complex to the nucleus where it affects translation of TGF-β associated genes [29]. Control group hVICs showed a moderate degree of nuclear translocation of activated SMAD2/3 complex, visible as bright nuclei on immunostaining (Fig 5A). This effect was entirely absent in FIB-media cells, although staining did show presence of the complex in the cytoplasm. The TGF-β treated cells had the greatest degree of nuclear translocation, with very bright nuclei present in all cells. FGF-2 treatment did not meaningfully alter the expression of *SMAD2* mRNA (Fig 5B and 5C), only showing a marginal increase in *SMAD3* expression on day 4. Only TGF-β significantly influenced mRNA expression of these genes, with a 2.4-fold decrease in *SMAD3* expression on day 2 (95% CI:

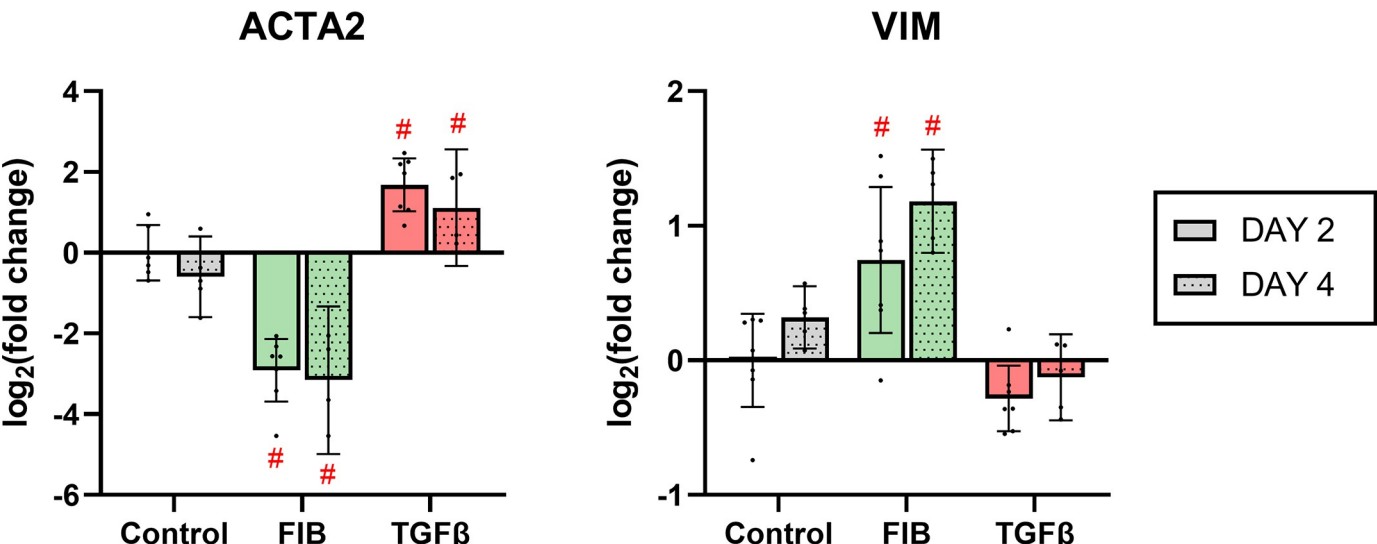

**Fig 4. FGF-2 downregulates ACTA2 expression.** RT-qPCR results of ACTA2 (α-SMA) and VIM (vimentin) expression. In a mixed effects ANOVA, experimental groups were compared to control at each time point. * p<0.05, #p<0.01, dots represent biological replicates (n = 7 on day 2, n = 5 on day 4), error bars represent 95% CI.

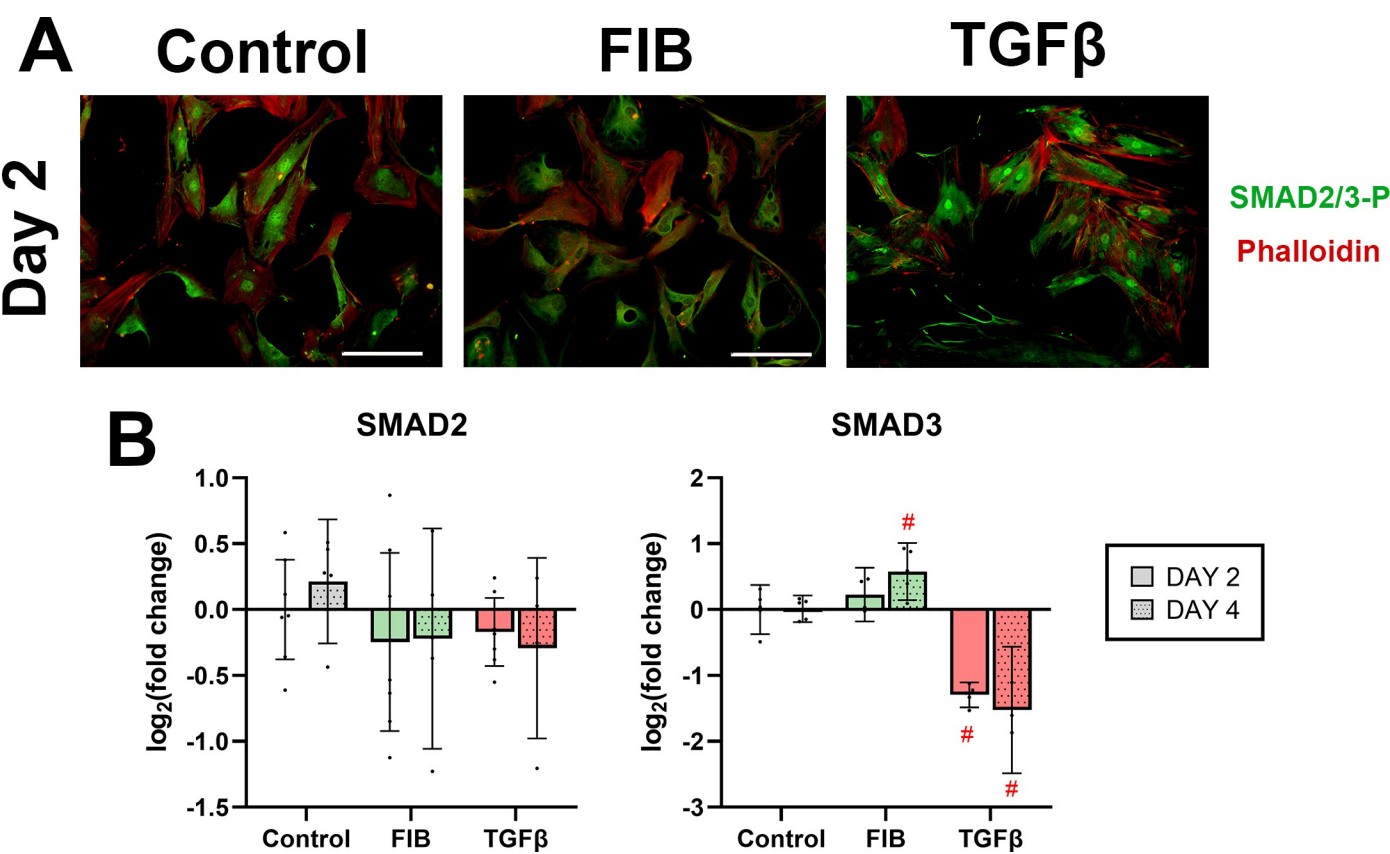

**Fig 5. FGF-2 reduces SMAD immunostaining, no effect on expression of SMAD genes.** (A) shows hVICs on day 2 stained for activated SMAD2/3 complex (green), counterstained with phalloidin (red), scale bars represent 100μm. Representative images of day 4 did not appreciably differ from day 2, and are available in the supplementary materials (S3). (B) shows RT-qPCR results of expression of SMAD2 and SMAD3 respectively at day 2 and 4. * $p < 0.05$, #$p < 0.01$, dots represent biological replicates (n = 7 on day 2, n = 5 on day 4), error bars represent 95% CI.

2.2–2.8, $p < 0.0001$) and a 2.8-fold decrease on day 4 (95% CI: 1.7–8.9, $p < 0.0001$). The data reaffirm the well-established canonical TGF-β signalling pathway in hVICs. FGF-2 does not, however, influence the expression of the *SMAD* genes.

## FGF-2 inhibits formation of focal adhesions

Focal adhesions are the protein complexes that join actin filaments of adjacent cells together to form a functional unit [30]. As these adhesions transmit tension forces from cell to cell, their presence is associated with the contractile nature of myofibroblasts. We used paxillin as a target for immunostaining to resolve focal adhesions in our hVIC cultures (Fig 6A). In the control group, focal adhesions appeared as brightly stained nodules along the cell margin, typically between 2 and 5μm in length. In the FGF-2 treated cells, the number of these foci was starkly reduced, and instead, paxillin staining appeared diffusely around the cytoplasm. By contrast, the TGF-β treated cells showed many mature focal adhesions formed along cell-cell junctions. We thresholded the images taken and counted the number of focal adhesions per cell (Fig 6B). There was no significant inter-time point difference. FGF-2 caused a significant decrease in the number of paxillin-positive focal adhesions, with a 2-fold decrease at day 2 and 4. Conversely, the TGF-β group showed a 2-fold increase at both time points. FGF-2 appears to inhibit formation of focal adhesions. TGF-β appears to promote their formation.

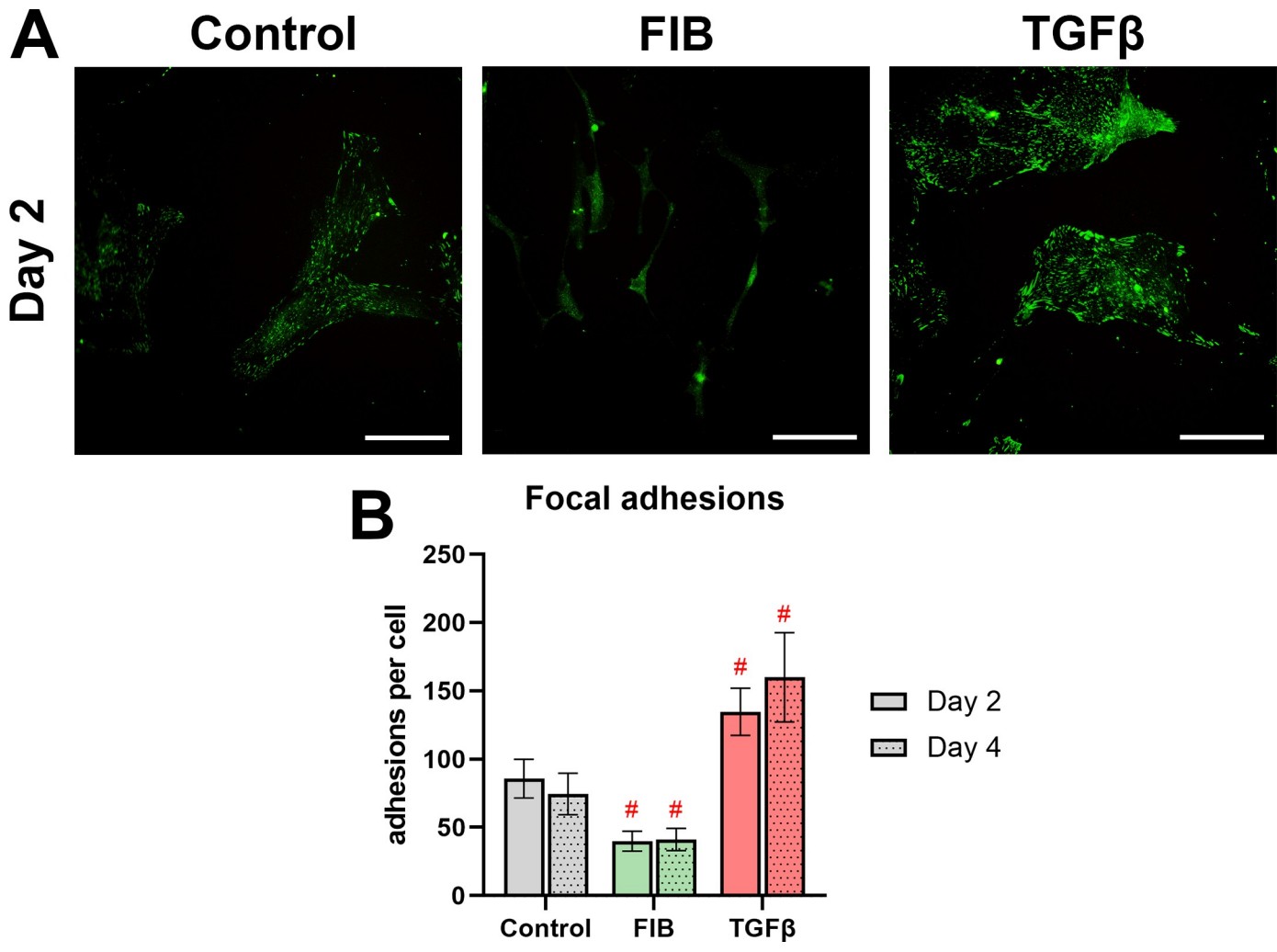

**Fig 6. FGF-2 inhibits focal adhesion formation.** (A) shows paxillin staining of hVICs on day 2, scale bars represent 100μm. Representative images of day 4 did not appreciably differ from day 2, and are available in the supplementary materials (S3). (B) is the number of mature focal adhesions per cell. n = 4 biological replicates were used, 10 images per replicate per group. Experimental groups were compared to control using a 2-way ANOVA. * p<0.05, #p<0.01, error bars represent 95% CI.

### FGF-2 inhibits nodular-type calcification, but did not affect osteoblastic differentiation of hVICs

Alizarin red staining was used to resolve calcification formed by hVICs on day 2 and 4 (Fig 7A). Specifically, we looked to distinguish the nodular-type calcification formed around apoptotic bodies from the diffuse calcification produced by osteoblast-like phenotype described in the literature [16, 20, 28]. All groups showed a significant increase in the number of calcific foci over the time course (Fig 7B, p<0.0001), though they did not significantly differ from each other on day 2. By day 4 the control group cells formed numerous calcific nodules. These nodules were small, measuring between 5 and 20 μm across, typical of apoptotic bodies. The FIB-media hVICs showed significantly fewer nodules compared to control (p<0.0001), with each well containing less than 20 resolvable calcifications. TGF-β treated cells on the other hand, formed considerably more nodules (p = 0.0001).

Interestingly, no evidence of diffuse osteoblast-like calcification was seen. Nor was there any change in genes associated with osteoblastic differentiation (Fig 8). We performed RT-

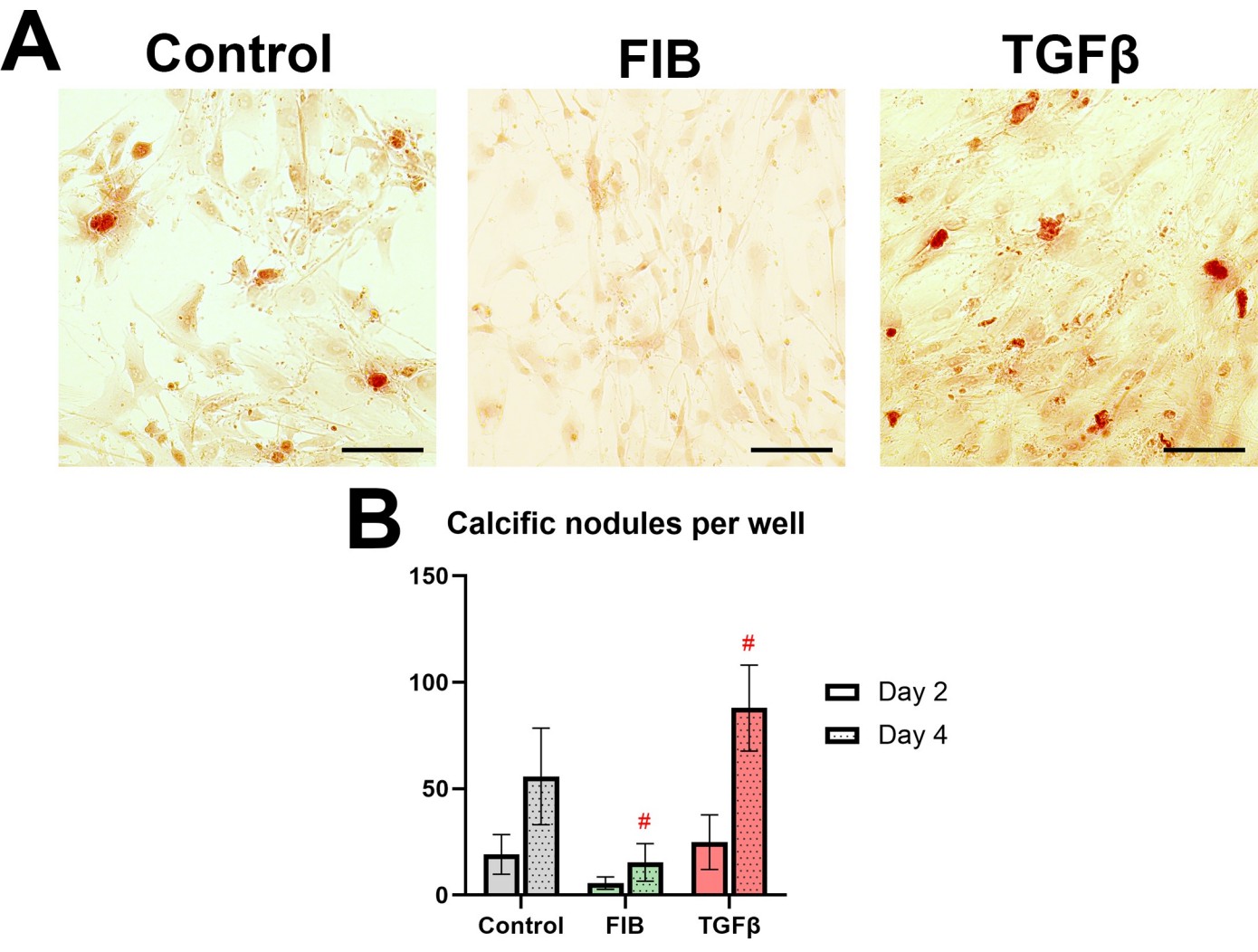

**Fig 7. FGF-2 inhibits nodular calcification.** (A) shows alizarin red stained hVICs on day 4. Scale bars represent 100μm. (B) shows the number of calcific nodules per well on day 2 and day 4, counted manually from n = 4 biological repeats, with 2 wells per group. A 2-way ANOVA compared the experimental groups with control at each time point. * p<0.05, #p<0.01, error bars represent 95% CI.

qPCR on *RUNX2*, the 'master transcription factor' of the osteoblast lineage and found no difference between groups (p = 0.9122). Levels of *OPN* and *OPG* were highly variable across replicates, and did not differ between groups (p = 0.2583 and 0.5175 for *OPN* and *OPG* respectively). Expression of *BGLAP* and *SP7* was below detectable levels. *CDH11*, a cadherin isoform associated with both osteoblasts and fibroblast more generally [31, 32], did not show any changes in expression under FGF-2 or TGF-β stimulation (p = 0.3112). Only biglycan (*BGN*), a bone-associated glycoprotein, demonstrated a significant upregulation in the TGF-β group (p = 0.0074 on day 2, p = 0.0006 on day 4), however this increase was small in magnitude. On the whole, expression of osteoblast-associated genes appeared unaffected by the treatments.

## FGF-2 inhibits expression of matrix synthesis, in opposition to TGF-β

Six matrix associated genes were assayed in this study at day 2 and day 4 (Fig 9). None of these genes showed a significant time-dependent change in expression. The overall effect

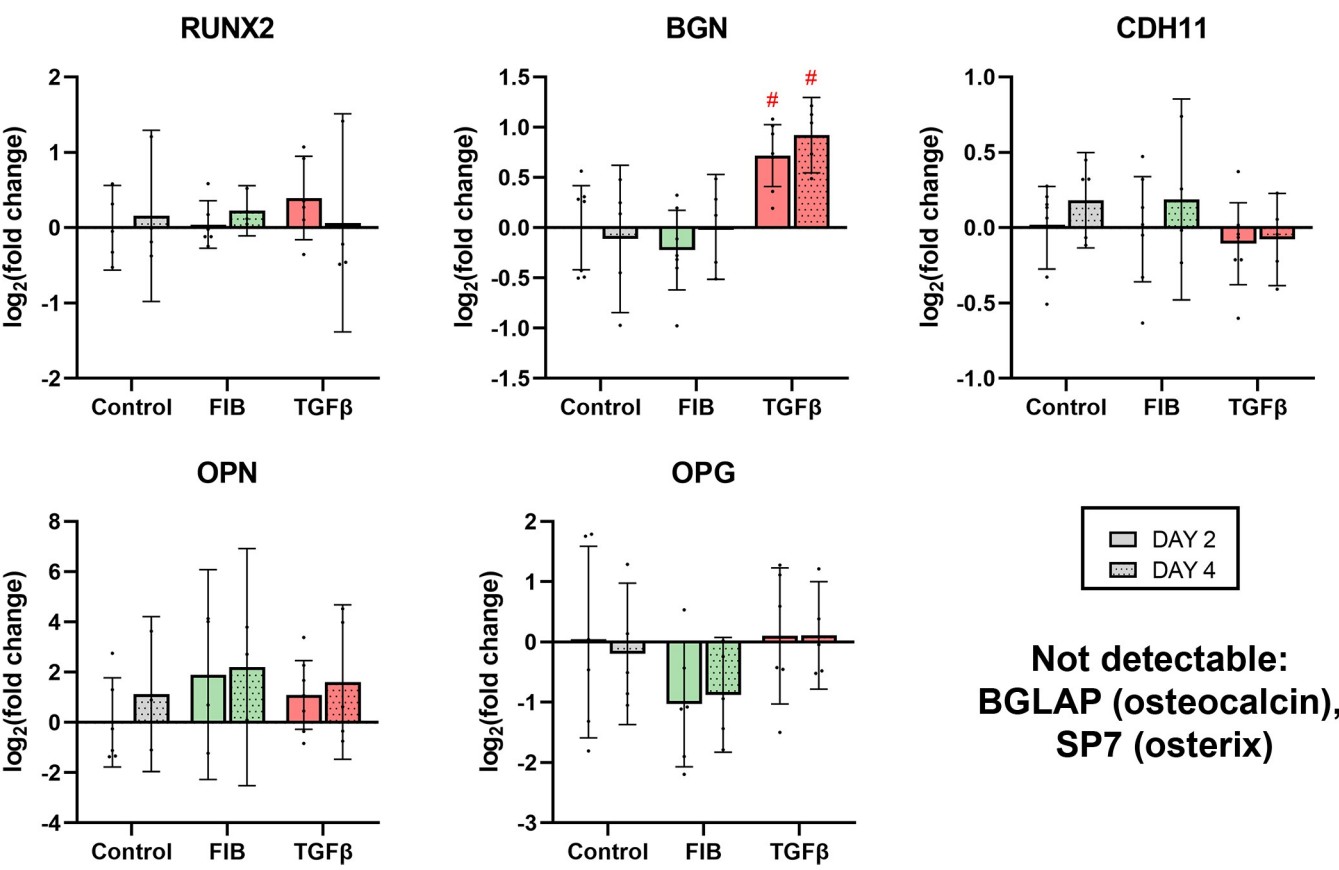

**Fig 8. FGF-2 has no effect on expression of osteoblast-associated genes.** Results of RT-qPCR on a panel of 7 osteoblast associated genes. Note that expression of BGLAP and SP7 was below the detectable level. * p<0.05, #p<0.01, dots represent biological replicates (n = 7 on day 2, n = 5 on day 4), error bars represent 95% CI.

of FGF-2 tended towards downregulation of structural matrix proteins and upregulation of proteinases, indicating a net matrix 'breakdown' effect. *COL1A1* encodes the major precursor to collagen 1, and FGF-2 caused a marginal but statistically significant downregulation of this gene on day 2. A striking difference was noted in the expression of elastin, where FGF-2 caused a 5.0-fold decrease in *ELN* mRNA (95% CI: 4.2–6.2) on day 2. *MMP1*, which codes for matrix metalloproteinase 1, an enzyme that digests collagen, was significantly upregulated at both timepoints, with a 9.8-fold and 7.0-fold increase on days 2 and 4 respectively. Changes to *MMP2* (type IV collagenase) and *TIMP1* (inhibitor of MMPs) did show some significant changes under FGF-2 stimulation, but these were marginal in magnitude. TGF-β treated hVICs in most instances demonstrated opposite effects on expression of matrix associated genes, i.e. the there was a tendency towards matrix synthesis. *COL1A1* and *COL3A1* expression was significantly increased on both days 2 and 4. Notably, levels of *ELN* mRNA were increased 3.7-fold on day 2 (95% CI: 2.5–3.7) and 5.0-fold on day 4 (95% CI: 3.3–6.7). One result that departed from the general theme of TGF-β stimulated matrix synthesis was the 2-fold increase in *MMP2* in the TGF-β group at both day 2 and 4. Overall, the RT-qPCR results show that FGF-2 stimulates a movement towards a matrix breakdown gene expression profile, while TGF-β stimulates a movement towards an increase in matrix associate genes.

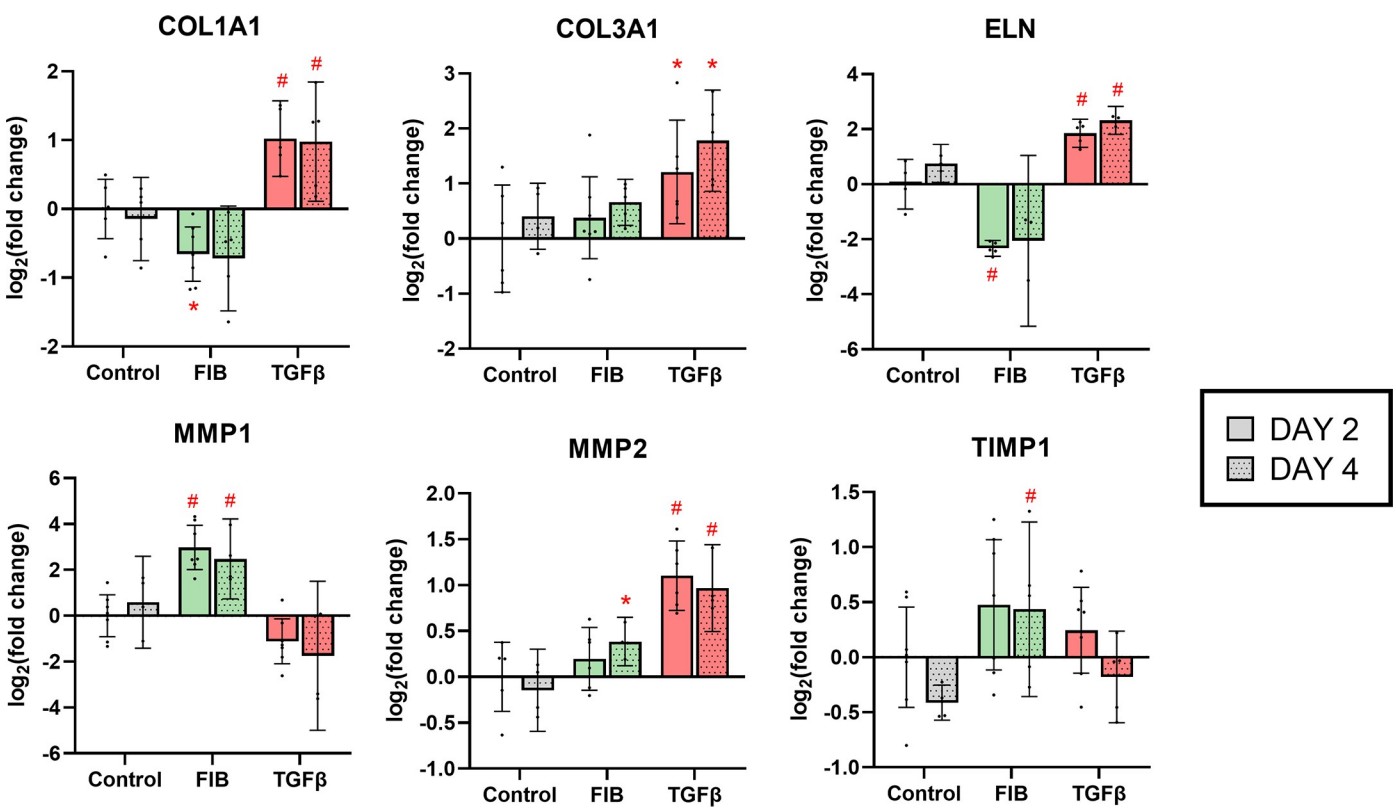

**Fig 9. FGF-2 promotes matrix breakdown expression pattern.** RT-qPCR results on a panel of matrix associated genes. * $p < 0.05$, #$p < 0.01$, dots represent biological replicates (n = 7 on day 2, n = 5 on day 4), error bars represent 95% CI.

## FGF-2 promotes expression of bone morphogenic proteins

Bone morphogenic proteins (BMPs) are members of the TGF-β family, influencing bone development. FGF-2 promoted an upregulation of *BMP2*, with a 5.0-fold increase on day 2 (95% CI: 2.1–8.0) and a 4.4-fold increase on day 4 (95% CI: 3.3–5.6) (Fig 10). There was a small but statistically significant increase in *BMP4* expression by day 4. TGF-β had the

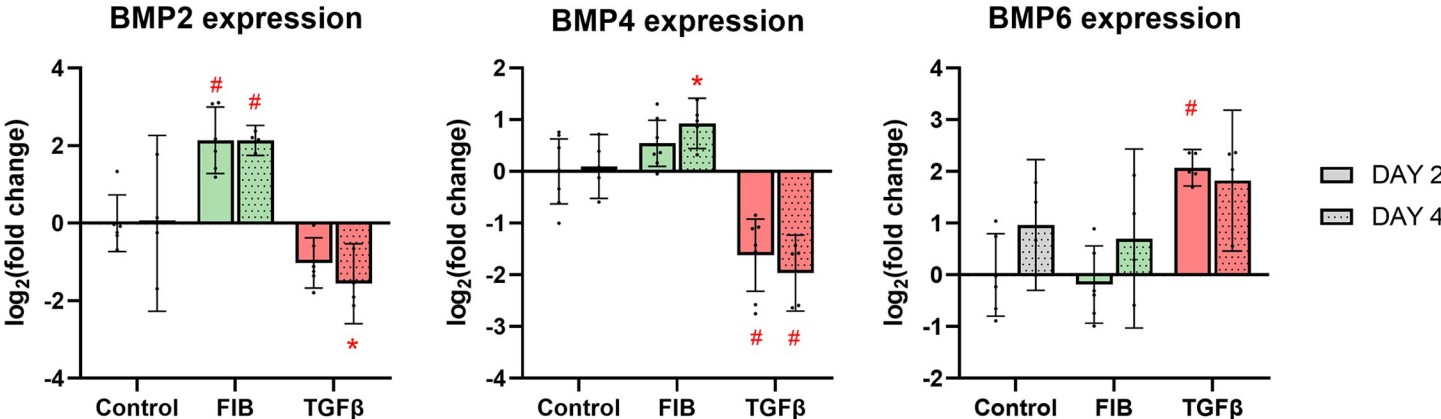

**Fig 10. FGF-2 upregulates BMP2 expression.** RT-qPCR results of BMPs 2, 4, and 6. In a mixed effects model, experimental groups were compared to control at each time point. * $p < 0.05$, #$p < 0.01$, dots represent biological replicates (n = 7 on day 2, n = 5 on day 4), error bars represent 95% CI.

opposite effect: *BMP2* was down regulated 2.7-fold on day 4 (95% CI: 1.5–13.1); and *BMP4* showed significant downregulation, with a 2.8-fold decrease on day 2 (95% CI: 2.0–4.6) and a 3.7-fold decrease on day 4 (95% CI: 2.5–6.8). Expression of *BMP6* was low across all groups, and demonstrated a high degree of variability between repeats. Despite this, TGF-β did induce a 2.1-fold upregulation of *BMP6* on day 2 (95% CI: 1.7–2.4).

### Scleraxis: A potential marker of hVIC myofibroblasts?

Scleraxis is a transcription factor typically associated with tendon development [33] and cardiac myofibroblasts [34]. We assayed the expression of *SCX* as part of our gene expression panel and found a marked degree of change in the experimental groups (Fig 11B). FGF-2 caused a 2.8- and 2.3-fold decrease on days 2 and 4 respectively. Conversely, TGF-β caused a 6.9-fold and 8.9-fold upregulation on day 2 and 4 respectively. Following this result, we used immuno-staining to identify presence of scleraxis in the hVICs. All experimental groups showed presence of scleraxis, localised to the nucleus (Fig 11A). However, there were no observed differences between the groups.

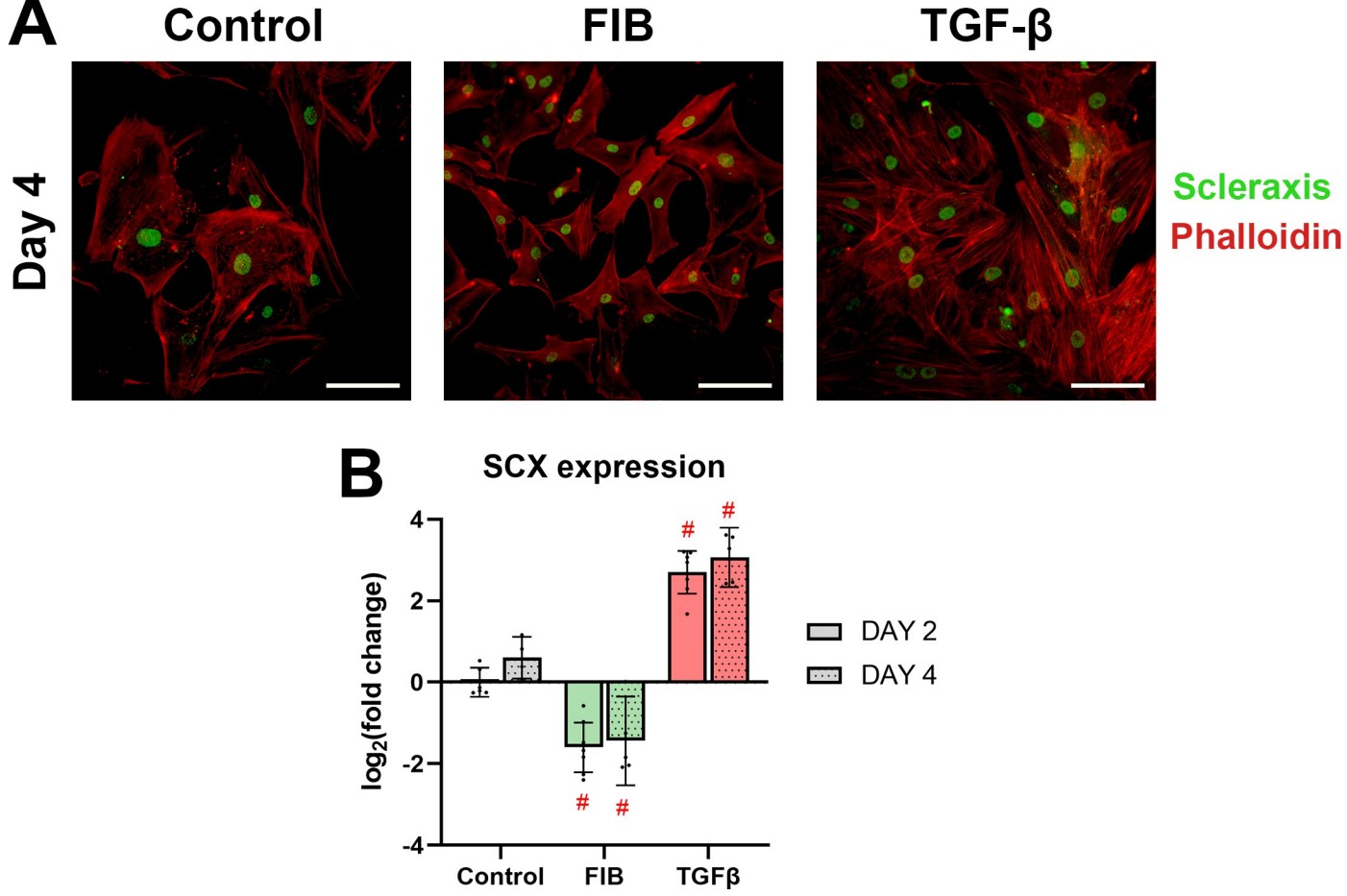

**Fig 11. Scleraxis, potential myofibroblast marker, is upregulated by TGF-β.** (A) shows representative images of hVICs stained for scleraxis (green) and phalloidin (red) on day 4. Representative images of day 2 did not appreciably differ from day 4, and are available in the supplementary materials (S3). Scale bars represent 100μm. (B) shows RT-qPCR results of SCX (scleraxis) gene expression. In a mixed effects model, experimental groups were compared to control at each time point. * $p < 0.05$, # $p < 0.01$, dots represent biological replicates (n = 7 on day 2, n = 5 on day 4), error bars represent 95% CI.

## Discussion

Our study found that FGF-2 had an anti-myofibroblast effect on hVICs isolated from patients with end-stage CAVD. Myofibroblasts are central to wound healing and growth of tissues [19]. However, excessive activation of VICs to myofibroblasts is associated with valve sclerosis and eventual calcification [16, 21]. In a study on healthy hVICs, Latif and colleagues described the use of FGF-2 to inhibit the myofibroblastic activation of VICs using a 'FIB' media formulation that contained 10ng/mL FGF-2 and a reduced serum content to lower endogenous TGF-β [24]. This study aimed to build on this work by demonstrating the inhibitory effect of FGF-2 on the myofibroblast activation of hVICs and formation of calcific nodules, with a focus on changes in gene expression. Latif et al's work showed that FGF-2 could *retain* hVICs in quiescence, but is not known if FGF-2 could *reverse* myofibroblastic activation in cells that were already activated. Unique to this study was the use of hVICs isolated from patients with end-stage CAVD; we wanted to determine if the therapeutic effect seen in healthy hVICs would remain intact in a population of diseased cells with a greater degree of myofibroblastic activation.

To examine the pro- or anti-myofibroblast effect of FGF-2, it was first necessary to define exactly what constitutes a 'myofibroblast'. Previous work in this area has used presence of α-SMA as a marker [17, 22, 24, 28], and while this is a well-validated approach, α-SMA is also present to a lesser degree in healthy valve tissue, particularly during growth and development [11]. Other researchers have used 'increased contractility' [22, 24], collagen deposition by hydroxyproline assay [22], desmin [17], or the activation of known TGF-β pathways as end-points [22, 35]. In our study, we used TGF-β stimulation as a positive control to build up a comprehensive picture of what a hVIC-derived myofibroblast looks like. We found that TGF-β treated VICs take on a distinct morphology: when cultured on plastic cells become very large and rounded, and lay down mature bands of α-SMA that terminate in focal adhesions. Focal adhesion formation was also increased by TGF-β. The gene expression profile of TGF-β treated hVICs was pro-matrix forming, with significant upregulations in genes coding collagen 1, collagen 3, and elastin. This effect has been demonstrated extensively in the literature, where TGF-β stimulation consistently upregulates matrix gene expression and deposition of VICs [35, 36]. *MMP2*, a matrix protease that breakdown collagen IV-rich basement membranes, was also upregulated. This has been shown in other cell types [37], and indeed is a mechanism behind cell invasion through basement membranes in tumour metastasis [38, 39].

Across most of our outcomes, FGF-2 induced an opposite effect to that caused by TGF-β stimulation. FGF-2 treated cells were more typical of quiescent fibroblasts in shape: they were of significantly smaller area and demonstrated a 'spindle-like' morphology. These findings were in line with Latif and colleagues [24]. A primary endpoint for aVICs, α-SMA expression, was reduced in the FGF-2 group, and enhanced in the TGF-β group. The gene expression profile showed reduced levels of matrix-forming proteins, and a large increase in *MMP1*, indicating a net degradative effect. This effect is seen in fibroblasts elsewhere: Kashpur and colleagues found similar reductions in matrix associated mRNAs coupled with upregulation of *MMP1* in dermal fibroblasts treated with FGF-2 [40]. How far these conclusions can translate to an *in vivo* setting remains unclear. In a gene expression analysis of normal vs stenotic human valves, Bossé and colleagues found that stenotic valves (i.e. valves with a high degree of myofibroblasts) showed a general upregulation in MMPs; *MMP1* and *MMP12* in particular [7].

Our findings regarding hVIC calcification are twofold: 1) FGF-2 inhibits the nodular-type calcification, and 2) no evidence of diffuse, osteoblast-like calcification occurred in our cultures. The TGF-β stimulated formation of nodules in porcine cell culture has been extensively described in the literature [22, 28, 41, 42], and mirrors what we saw with TGF-β treatment of our hVICs. Both Cushing et al. and Gao et al. describe a similar reduction in calcification

under FGF-2 stimulation [22, 25]. However, an interesting departure from the literature is the apparent absence of change in cadherin 11 gene expression in our study. Cadherin-11 is described by Hutcheson and colleagues as necessary in the formation of 'dystrophic' nodules in porcine cells [28]–an effect absent here. It is widely accepted that TGF-β treatment does induce osteoblastic differentiation of VICs, including the upregulation of osteoblast associated genes [16, 43, 44]. Our study did not reveal marked changes in osteoblast related genes upon FGF-2 or TGF-β stimulation, nor did we find evidence of ossification typical of osteoblast-like cells. Gao and colleagues did find FGF-2 mediated downregulations of osteogenic genes *RUNX2*, *OPN*, and *SP7*, though they used osteogenic medium [25]. The only finding suggestive of osteoblast differentiation was TGF-β stimulated upregulation of biglycan, an effect described by Song and colleagues as a crucial part of 'pre-osteogenic programming' in hVICs [45]. It is well established that hVICs are a heterogenous cell population, with various authors recognising up to five distinct phenotypes [16, 21]. It is unclear from this literature if all VICs undergo osteoblast differentiation, or if only certain subtypes are susceptible. It is unclear from our study if our culture conditions could induce osteogenesis in hVICs, or if the cells we isolated from diseased valves were somehow less responsive to osteoblastic differentiation than porcine or healthy human cells used by other researchers. The osteoblastic VICs appear to arise from quiescent VICs independent of the myofibroblastic lineage [46], and it is possible that the cells used here were committed myofibroblasts. FGF-2 stimulation upregulated the expression of BMPs 2 and 4, while TGF-β treatment caused a downregulation of the same. This effect has been described [44], and may be ascribed to the negative feedback TGF-β exerts on expression of TGF-β family of signallers, of which the BMPs are members [47]. Our finding is in opposition to those of Gao and colleagues, who describe the down-regulation of BMP2 in response to FGF-2 stimulation via the Notch1 pathway, though the FGF-2 in this case was added to osteogenic medium [25]. The disparate findings may represent the effect of FGF-2 on the two distinct modes of VIC calcification (nodular and osteoblastic), and Gao et al.'s down-regulation of osteoblast genes and calcification occurred via Notch1-mediated obVIC inhibition—an effect not seen in our study.

We found FGF-2 to slightly improve viability on alamarBlue assay. Both the absolute absorbance values, and the rate of increase of said values were raised in the FIB group, and did not differ from control in the TGF-β group. Other researchers have found similar [25] or the opposing effects [24]. Our study has used insulin as a mitotic agent in the FIB medium, and this addition may explain our findings here. Lam and colleagues, who used the same insulin concentration in their FGF-2 supplemented media, also found increases in proliferation [48].

The canonical TGF-β signalling pathway was intact in our hVIC culture, evidenced by the localisation of the activated SMAD2/3 complex to the nucleus upon immunostaining. We found that FIB media inhibited this localisation, as much less nuclear staining was seen compared to control, in line with previous studies on porcine cells [22, 23]. It is unclear if the low SMAD2/3 localisation was due to the FGF-2 alone or the low endogenous TGF-β in the FIB media, though Cushing et al.'s findings in porcine VICs support FGF-2's inhibitory action [22].

Scleraxis emerged as a potential marker of myofibroblastic VICs. Scleraxis is known to be a critical factor in conversion of cardiac fibroblasts to myofibroblasts via the canonical TGF-β signalling pathway [34, 49], but it is unclear from this study if this upregulation represents a true *in vivo* effect in valve cells specifically. Scleraxis is traditionally thought of as a tendon marker [33], but embryological studies in mice show scleraxis as a key transcription factor for heart valve development [50], and its upregulation has been ascribed to canonical TGF-β2 signalling [51].

In all, FGF-2 promoted a quiescent-like phenotype in hVICs isolated from calcific valves, while TGF-β promoted a myofibroblastic 'aVIC' phenotype. Our findings are summarised in Fig 12.

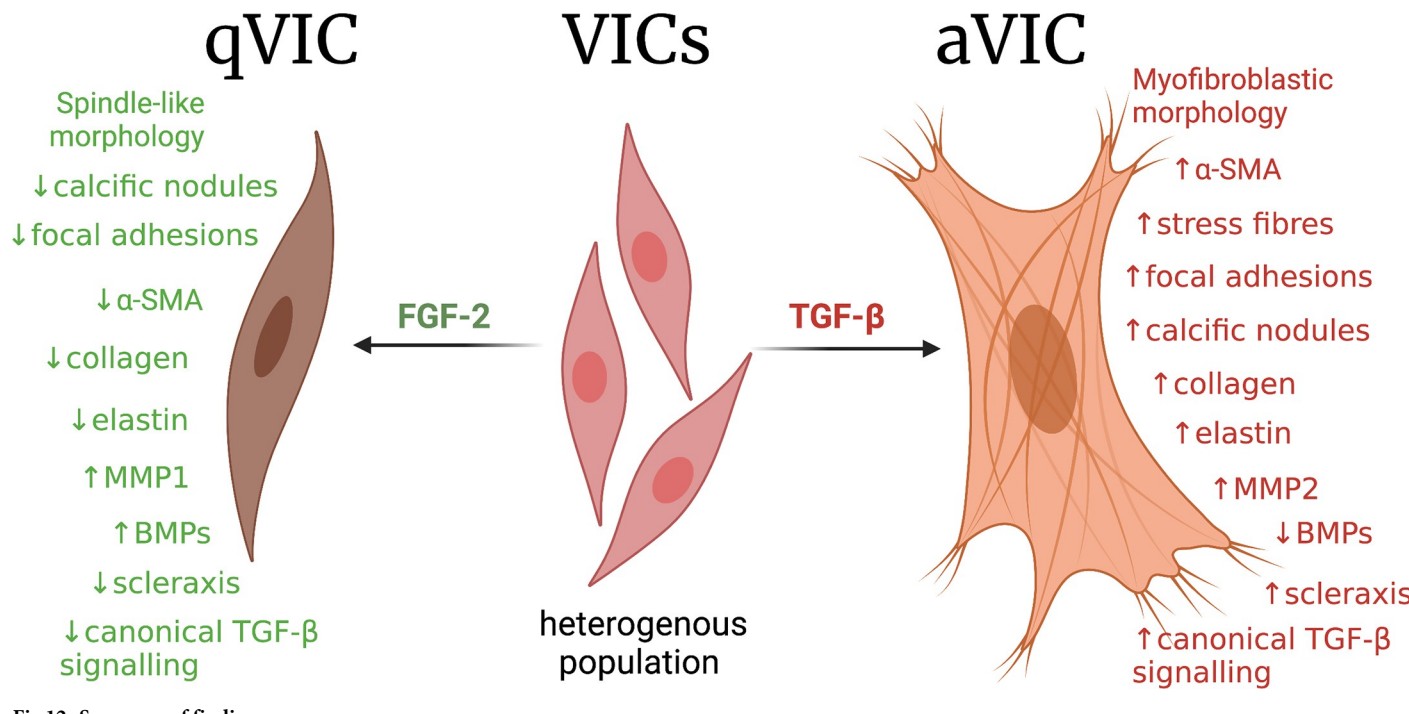

**Fig 12. Summary of findings.**

## Conclusion

Our findings suggest that FGF-2 opposes the formation of aVICs in an *in vitro* setting. We assayed cells from patients with end stage valve disease, i.e. cells with a high degree of inherent activation, and found that outcomes associated with myofibroblastic activation were returned to 'quiescent-type' levels by FGF-2. We used the well-validated aVIC marker, α-SMA, as well as a panel of gene expression analyses to build a comprehensive picture of 'myofibroblastic activation' and noticed that FGF-2 pushed cells in the opposite direction. Pro-quiescent observations include the downregulation of ECM production and inhibition of calcific nodule formation. FGF-2 showed an *in vitro* therapeutic effect, returning cells from an apparent diseased state to a quiescent state.

## Supporting information

**S1 Table. List of probes assayed in RT-qPCR.**
(DOCX)

**S2 Table. Measures of RNA purity and integrity.**
(DOCX)

**S1 Fig.** Immunocytochemistry from alternate timepoints (day 4 SMAD2/3, day 4 paxillin, and day 2 scleraxis) A) shows hVICs on day 4 stained for activated SMAD2/3 complex (green), counterstained with phalloidin (red), B) shows hVICs on day 4 stained for activated paxillin (green), C) shows hVICs on day 2 stained for activated scleraxis (green), counterstained with phalloidin (red). Scale bars represent 100μm.
(TIF)

## Acknowledgments

The University of Auckland's Bone and Joint Research Group were immensely helpful throughout the course of this research.

## Author Contributions

**Conceptualization:** Marcus Ground, Karen Callon.

**Data curation:** Marcus Ground.

**Formal analysis:** Marcus Ground.

**Funding acquisition:** Marcus Ground, Steve Waqanivavalagi.

**Investigation:** Marcus Ground, Young-Eun Park, Karen Callon.

**Methodology:** Marcus Ground, Young-Eun Park, Karen Callon.

**Project administration:** Marcus Ground.

**Resources:** Marcus Ground.

**Supervision:** Robert Walker, Paget Milsom, Jillian Cornish.

**Validation:** Marcus Ground, Young-Eun Park.

**Writing – original draft:** Marcus Ground.

**Writing – review & editing:** Marcus Ground.

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
