## [Decision Letter · Decision Letter 0]

21 Mar 2022

PONE-D-22-04010Fibroblast growth factor 2 inhibits myofibroblastic activation of valvular interstitial cellsPLOS ONE

Dear Dr. Ground,

Thank you for submitting your manuscript to PLOS ONE. After careful consideration, we feel that it has merit but does not fully meet PLOS ONE’s publication criteria as it currently stands. Therefore, we invite you to submit a revised version of the manuscript that addresses the points raised during the review process.

Please clarify the new information being reported in the current study as related to previous reports where appropriate.  Additional review comments are listed below.

We look forward to receiving your revised manuscript.

Kind regards,

Katherine Yutzey, PhD

Academic Editor

PLOS ONE

Journal Requirements:

https://journals.plos.org/plosone/s/file?id=ba62/PLOSOne_formatting_sample_title_auSimilarity Check/iThenticate Results (28%)thors_affiliations.pdf.

4. Please ensure that you refer to Figure 10 in your text as, if accepted, production will need this reference to link the reader to the figure.

Reviewers' comments:

Reviewer's Responses to Questions

**Comments to the Author**

1. Is the manuscript technically sound, and do the data support the conclusions?

Reviewer #1: Partly

Reviewer #2: Partly

2. Has the statistical analysis been performed appropriately and rigorously? 

Reviewer #1: Yes

Reviewer #2: No

3. Have the authors made all data underlying the findings in their manuscript fully available?

Reviewer #1: Yes

Reviewer #2: No

4. Is the manuscript presented in an intelligible fashion and written in standard English?

Reviewer #1: Yes

Reviewer #2: Yes

5. Review Comments to the Author

Reviewer #1: This manuscript compares VIC phenotype upon treatment with FGF2 or TGFb1, with a specific focus on myofibroblastic and osteogenic markers. This is possibly the most clearly written paper that this reviewer has seen in a long time; it was a pleasant read. That said, this reviewer does have several concerns:

1) Some of the findings in this paper have been previously published by others – e.g., VIC morphology measures with FGF2 or TGFb1 treatment, aSMA upon FGF2 or TGFb1 treatment. Several of these previous works are indeed referenced by the authors, but many of the authors’ findings – which mirror those previous studies – do seem a bit redundant in the context of already-published work. Is the unique contribution of the current study that the VICs were from diseased valves? If so, that aspect should be better emphasized to differentiate this study from earlier ones. If not, it would be useful to perhaps place some of the ‘repeat’ data in the supplement and focus the main paper on measures that have not been previously reported.

2) The authors state that “FGF-2 interferes with the nuclear translocation of the [Smad 2/3] complex” because translocation was lower in the FIB condition than in the Control. However, as noted by the authors, the Control condition (with 10% FBS) already has more TGFb1 in it than the FIB condition. So, it seems likely that Smad 2/3 translocation may be lower in FIB media simply because there is less TGFb1 present than in the Control media? In other words, less Smad 2/3 translocation can just indicate less TGFb1 to stimulate it, not an active inhibitory role of FGF2.

3) The authors used hVICs from 40 different donors. Were those cell populations pooled together to perform the experiments? Or were different experiments performed with different donors? Additional information is needed on how many donors are represented in each experiment.

4) There are many places where it sounds like the authors have treated with FGF2 and then treated those cultures with TGFb1 (which, as an aside, would be an interesting future experiment). For example, the abstract indicates treatment with “FGF2 and TGFb1”, and there are multiple variations of the conclusion that “FGF-2 inhibits the nodular-type calcification that results from TGF-β stimulation”. Other statements such as “FGF-2 reduced the calcification potential of VICs” (in the abstract) also made this reviewer think that this paper was going to be about an active calcification stimulus being added after FGF2.

5) Another VIC/FGF2 study just came out last month (PMID: 35074856), likely after the authors had already finalized this manuscript. This work is distinct, but should be cited/discussed as appropriate in the current manuscript.

6) A table containing patient demographics (age, sex, other relevant info if available) should be included.

Reviewer #2: In this study human aortic valve interstitial cells isolated from diseased valves were used for in vitro studies on how FGF-2 affects cell progression toward disease. The authors found that FGF-2 added to culture medium inhibited the activation of interstitial cells to myofibroblasts and decreased the formation of calcified nodules. Several comments/questions on the manuscript are below:

1) How were the 2 and 4 days time points chosen? It is not clear why there are two different time points, and in some figures it is not clear if the image analysis is for day 2 or day 4 images (i.e. Fig 2, Fig 3). Why are there no day 4 images in Fig 5 and Fig 6?

2) In Fig 11 the control image seems to be at a different magnification.

3) Figure legends can be improved. For all legends there is no mention of the number of replicates (n = XX). In image analysis graphs there is no mention of the number of images measured or cells/image measured. Please add this information. In the calcific nodule counting methods there is no mention of the number of replicates or number of wells analyzed.

4) Statistical analysis and annotation on figures is not clear. I believe a two way anova was performed, so in Fig 2, for example, are annotations denoting statistical significance relative to control or other treatments? If there is statistical significance between different treatments that should be indicated on the figures. I have similar questions about the other figures as well, statistical significance between treatments should be tested.

5) Page 16: This, coupled with the apparent absence of diffuse calcification suggests that FGF-2 inhibits calcification of hVICs with a mechanism independent of osteoblast differentiation.

I find this statement confusing and a bit broad. These cells were isolated from calcified valves that had already moved past osteoblastic differentiation and into large calcified nodules. It seems that pro-osteoblastic RUNX2 gene expression would normally happen earlier in the disease process. End stage valve disease cells may be less "plastic" and responsive to signaling. Could this statement be softened with some qualifiers about the initial state of the cells that were isolated, or can the authors add this point to the discussion?

6. PLOS authors have the option to publish the peer review history of their article (what does this mean?). If published, this will include your full peer review and any attached files.

Reviewer #1: No

Reviewer #2: **Yes: **Gretchen Mahler

---

## [Author Response · Author response to Decision Letter 0]

26 May 2022

Response to reviewers

Journal Requirements:

https://journals.plos.org/plosone/s/file?id=ba62/PLOSOne_formatting_sample_title_auSimilarity Check/iThenticate Results (28%)thors_affiliations.pdf.

We are eager to adhere to the style guide, and we think we have done this. If there are any areas where we have failed, please let us know

Our cover letter now includes:

“We were granted ethical approval to collect human tissue from patients at Auckland City Hospital. Māori, the indigenous population of New Zealand, view data as the ‘shadow’ of their tissue, and as such, data availability and data governance are very tightly controlled. Our ethical approval was granted by the Auckland Health Research Ethics Committee (AHREC), who have stipulated that all raw data must be stored securely on the premises of the University of Auckland, under the guardianship of either Marcus Ground or Professor Jillian Cornish. The key restriction that AHREC imposes on us is that the data must be housed in Auckland under our care. We are able to, however, share this data with other researchers upon any reasonable request. Data requests can be made to Marcus Ground (groma788@student.otago.ac.nz), Professor Jillian Cornish (j.cornish@auckland.ac.nz) or to AHREC directly at humanethics@auckland.ac.nz”

We hope you understand that the treatment of data as both personal and sacred by Maori in New Zealand means that we have to be ‘guardians’ of the data itself. This means we must keep it securely on our password-protected servers or locked in cabinets at the University of Auckland, per our AHREC approval. We are able to share data upon reasonable request.

We have removed the reference to data not presented here. We have instead added both time points to figure 5

4. Please ensure that you refer to Figure 10 in your text as, if accepted, production will need this reference to link the reader to the figure.

We have now included reference to figure 10

Reviewer #1: This manuscript compares VIC phenotype upon treatment with FGF2 or TGFb1, with a specific focus on myofibroblastic and osteogenic markers. This is possibly the most clearly written paper that this reviewer has seen in a long time; it was a pleasant read. That said, this reviewer does have several concerns:

1) Some of the findings in this paper have been previously published by others – e.g., VIC morphology measures with FGF2 or TGFb1 treatment, aSMA upon FGF2 or TGFb1 treatment. Several of these previous works are indeed referenced by the authors, but many of the authors’ findings – which mirror those previous studies – do seem a bit redundant in the context of already-published work. Is the unique contribution of the current study that the VICs were from diseased valves? If so, that aspect should be better emphasized to differentiate this study from earlier ones. If not, it would be useful to perhaps place some of the ‘repeat’ data in the supplement and focus the main paper on measures that have not been previously reported.

We acknowledge that we have used many of the same methodologies as previous researchers. However, to our knowledge, this is the first use of FGF-2 on cells explicitly from diseased valves, and is also the first that demonstrates ‘reversal’ of already activated cells. The cell population used by Latif et al. were from healthy donors, and so were mostly quiescent. Whereas the cell population used here is already highly myofibroblastic (We refer you the number of red-stained cells in the day 2 control group of Figure 3A). In any case, we have made the following changes to address your concern:

• The introduction has been amended to highlight the fact that this study is the first to expand previous FGF-2 findings onto diseased cells

• The first paragraph of the discussion has been amended to highlight what makes this study different

2) The authors state that “FGF-2 interferes with the nuclear translocation of the [Smad 2/3] complex” because translocation was lower in the FIB condition than in the Control. However, as noted by the authors, the Control condition (with 10% FBS) already has more TGFb1 in it than the FIB condition. So, it seems likely that Smad 2/3 translocation may be lower in FIB media simply because there is less TGFb1 present than in the Control media? In other words, less Smad 2/3 translocation can just indicate less TGFb1 to stimulate it, not an active inhibitory role of FGF2.

We agree we have overreached with this conclusion. The SMAD2/3 section of the results has been amended to remove suggestion that FGF-2 treatment alone caused this change, and have added a small sentence in the SMAD2/3 paragraph of the discussion. We direct you to Cushing et al. 2008 who showed thar FGF-2 does inhibit SMAD2/3 signalling in healthy pig cells: (doi: 10.1096/fj.07-087627)

3) The authors used hVICs from 40 different donors. Were those cell populations pooled together to perform the experiments? Or were different experiments performed with different donors? Additional information is needed on how many donors are represented in each experiment.

We did not pool donors. The ‘experimental conditions’ section of the methods has been amended to reflect this.

4) There are many places where it sounds like the authors have treated with FGF2 and then treated those cultures with TGFb1 (which, as an aside, would be an interesting future experiment). For example, the abstract indicates treatment with “FGF2 and TGFb1”, and there are multiple variations of the conclusion that “FGF-2 inhibits the nodular-type calcification that results from TGF-β stimulation”. Other statements such as “FGF-2 reduced the calcification potential of VICs” (in the abstract) also made this reviewer think that this paper was going to be about an active calcification stimulus being added after FGF2.

We agree that in several instances we have incorrectly made it sounds like FGF-2 and TGF-β treatments are conflated. We have amended the following points:

• The final sentence of the ‘FGF-2 inhibits formation of focal adhesions’ section of the results has been tweaked to remove the notion that somehow the FGF-2 worked against TGF-β in the same experiment

• The final sentence of the ‘FGF-2 inhibits expression of matrix synthesis, in opposition to TGF-β’ section of the results has been tweaked to remove the notion that somehow the FGF-2 worked against TGF-β in the same experiment

• The sentence in the conclusion that you mentioned was misleading has been amended to reflect the FGF-2 mediated changes only. And the TGF-β related changes have been expanded in the following sentence 

• The sentence referring to the up/down regulation of BMPs has been amended to separate out the FGF-2 and TGF-β arms of the study

• The sentence about FGF-2 and SMAD2/3 staining: the word ‘interfered’ was changed to ‘inhibited’ to remove the idea that somehow the two treatments are happening at once

• The conclusion has been amended to remove the notion that FGF-2 inhibited TGF-β effects on cells

That being said, our claims about FGF-2’s effects (inhibition of calcification etc.) are made when we compared FGF-2 to control (no treatment). I.e. if we removed the TGF-β group from the analysis entirely, then we would still see the reduction of calcification on alizarin red staining, etc. FGF-2 did in fact cause a >two thirds reduction in nodular calcifications when compared to control on day 4. 

5) Another VIC/FGF2 study just came out last month (PMID: 35074856), likely after the authors had already finalized this manuscript. This work is distinct, but should be cited/discussed as appropriate in the current manuscript.

We thank you for directing us to this paper - indeed we had mostly finished our manuscript. We absolutely agree that it should be discussed. To achieve this, we have:

- Made brief mention of it in the introduction

- Made brief mention of the anti-calcific effects in the discussion

- Added Gao’s BMP2 findings to the BMP2 section of the discussion. We have highlighted Gao’s use of osteogenic media, and how the disparate findings may be due to this

- Added the proliferation findings to the alamarBlue section of the discussion

6) A table containing patient demographics (age, sex, other relevant info if available) should be included.

Due to the nature of our ethics approval we aren’t able to share demographic information about our tissue donors. Our consenting nurses did record this information, but it is not available to researchers. All we know is that the donors were undergoing valve replacement surgery.

Reviewer #2: In this study human aortic valve interstitial cells isolated from diseased valves were used for in vitro studies on how FGF-2 affects cell progression toward disease. The authors found that FGF-2 added to culture medium inhibited the activation of interstitial cells to myofibroblasts and decreased the formation of calcified nodules. Several comments/questions on the manuscript are below:

1) How were the 2 and 4 days time points chosen? It is not clear why there are two different time points, and in some figures it is not clear if the image analysis is for day 2 or day 4 images (i.e. Fig 2, Fig 3). Why are there no day 4 images in Fig 5 and Fig 6?

With respect to our myofibroblast endpoints (α-SMA, cell morphology, etc.), we wanted to use two timepoints to determine if there was some directionality in the changes we saw (i.e. was the FGF-2 effect transient? Was is static, but sustained? Did it increase over the time course?). 

We chose to use day 2 and day 4 as timepoints as this, in our experience, is the time course in which hVICs undergo visible changes in their phenotype, but importantly, occurs prior to the cells become confluent (which generally occurs by day 6). Using low seeding densities (<3000 cells/cm2) to extend the growth phase, in our experience, can result in non-proliferating cultures.

We also drew on the literature. Other researchers have used VICs over a similar time course:

• Cushing et al. (doi: 10.1096/fj.07-087627) examined FGF-2 stimulation of porcine VICs over 2 days

• Monzack et al. (PMID: 21863660) examined osteoblast potential of VICs over 5 days

• Walker et al. (PMID: 15217906) treated VICs with TGF-β over the course of 2 days

• Gao et al. (PMID 35074856) used VICs over the course of 2 days

We agree there is inconsistency in the presentation of time points. To address your concerns:

• Figure 3B-E (the morphology analysis) is the result of pooling the day 2 and day 4 timepoints. We have updated the caption to reflect this. We pooled these data because in all morphology metrics, the 2-way ANOVA showed no significant differences between the time points, and we didn’t want to overload the graph with information.

• Figure 5 (Smads). We have now included both rtPCR time points on the graph. We did not see appreciable differences in staining so we only wanted to present one set of images. Day 4 staining is now included in supplementary figure 3.

• Figure 6 (paxillin). We did not see appreciable differences in the day 2 and day 4 paxillin staining and so we wanted to only present one set of images. We have now included day 4 staining in the supplementary figure 3

• Figure 11 (scx). We chose to stain for scleraxis because we saw gene expression changes at both time points. We did not see appreciable differences in the day 2 and day 4 scleraxis staining and so we wanted to only present one set of images. We have now included day 2 staining in the supplementary figure 3

2) In Fig 11 the control image seems to be at a different magnification.

We agree it looks fishy, but we promise the magnifications are the same. In our experience, there is a great degree of pleomorphism with the hVICs - particularly those in the control or TGF-β groups. For reference, have a look at these two images taken at the same magnification:

3) Figure legends can be improved. For all legends there is no mention of the number of replicates (n = XX). In image analysis graphs there is no mention of the number of images measured or cells/image measured. Please add this information. In the calcific nodule counting methods there is no mention of the number of replicates or number of wells analyzed.

We agree with your concerns about the lack of ‘N’s. To make this more transparent, we’ve done the following:

• Sample sizes for the various image analyses has been added to the ‘immunocytochemistry’ and the ‘alizarin red’ sections of the methods

• Captions from figures 2-11 have been amended to include replicates and well numbers where relevant

4) Statistical analysis and annotation on figures is not clear. I believe a two way anova was performed, so in Fig 2, for example, are annotations denoting statistical significance relative to control or other treatments? If there is statistical significance between different treatments that should be indicated on the figures. I have similar questions about the other figures as well, statistical significance between treatments should be tested.

We agree that our reporting of stats is lacking. To make our statistics more transparent, we have made the following changes:

• More explicit reference to the 2-way ANOVA has been added to the ‘statistics’ section. Specifically we have described explicitly what ‘image analysis’ is, and what values went into the gene expression 2-way ANOVAs

• We did not seek to measure differences between the two experimental groups - only the difference between each experimental group and the control group. We have explicitly stated this in the ‘statistics’ section

• The ‘alamarBlue’ section of the methods has been altered to make the comparisons more explicit.

• Captions have been amended where relevant

5) Page 16: This, coupled with the apparent absence of diffuse calcification suggests that FGF-2 inhibits calcification of hVICs with a mechanism independent of osteoblast differentiation.

I find this statement confusing and a bit broad. These cells were isolated from calcified valves that had already moved past osteoblastic differentiation and into large calcified nodules. It seems that pro-osteoblastic RUNX2 gene expression would normally happen earlier in the disease process. End stage valve disease cells may be less "plastic" and responsive to signaling. Could this statement be softened with some qualifiers about the initial state of the cells that were isolated, or can the authors add this point to the discussion?

We agree that this statement was over-reaching. It has been removed from the ‘results’ section. To provide further clarification, we have amended the fourth paragraph of our discussion to clarify that our diseased cells may have reduced ability to undergo osteoblastic differentiation. We have included reference to Monzack 2011 (PMID 21863660), who describes how obVICs may arise from qVICs independent of the myofibroblast lineage

---

## [Decision Letter · Decision Letter 1]

7 Jun 2022

Fibroblast growth factor 2 inhibits myofibroblastic activation of valvular interstitial cells

PONE-D-22-04010R1

Dear Dr. Ground,

We’re pleased to inform you that your manuscript has been judged scientifically suitable for publication and will be formally accepted for publication once it meets all outstanding technical requirements.

Kind regards,

Katherine Yutzey, PhD

Academic Editor

PLOS ONE

Additional Editor Comments (optional):

Reviewers' comments:

Reviewer's Responses to Questions

**Comments to the Author**

1. If the authors have adequately addressed your comments raised in a previous round of review and you feel that this manuscript is now acceptable for publication, you may indicate that here to bypass the “Comments to the Author” section, enter your conflict of interest statement in the “Confidential to Editor” section, and submit your "Accept" recommendation.

Reviewer #1: All comments have been addressed

Reviewer #2: All comments have been addressed

2. Is the manuscript technically sound, and do the data support the conclusions?

Reviewer #1: Yes

Reviewer #2: Yes

3. Has the statistical analysis been performed appropriately and rigorously? 

Reviewer #1: Yes

Reviewer #2: Yes

4. Have the authors made all data underlying the findings in their manuscript fully available?

Reviewer #1: (No Response)

Reviewer #2: No

5. Is the manuscript presented in an intelligible fashion and written in standard English?

Reviewer #1: Yes

Reviewer #2: Yes

6. Review Comments to the Author

Reviewer #1: (No Response)

Reviewer #2: My comments have been addressed, but it may still be helpful to better annotate statistical significance on the figures (i.e. add a bracket between statistically significant treatments)

7. PLOS authors have the option to publish the peer review history of their article (what does this mean?). If published, this will include your full peer review and any attached files.

Reviewer #1: No

Reviewer #2: **Yes: **Gretchen Mahler

---

## [Editor Report · Acceptance letter]

9 Jun 2022

PONE-D-22-04010R1 

Fibroblast growth factor 2 inhibits myofibroblastic activation of valvular interstitial cells 

Dear Dr. Ground:

I'm pleased to inform you that your manuscript has been deemed suitable for publication in PLOS ONE. Congratulations! Your manuscript is now with our production department. 

Kind regards, 

on behalf of

Dr Katherine Yutzey 

Academic Editor

PLOS ONE